# Diagnostic Effect of Consultation Referral from Gastroenterologists to Generalists in Patients with Undiagnosed Chronic Abdominal Pain: A Retrospective Study

**DOI:** 10.3390/healthcare9091150

**Published:** 2021-09-03

**Authors:** Ren Kawamura, Yukinori Harada, Taro Shimizu

**Affiliations:** Department of Diagnostic and Generalist Medicine, Dokkyo Medical University Hospital, Tochigi 321-0293, Japan; aetenanorunara.ren@gmail.com (R.K.); yuki.gym23@gmail.com (Y.H.)

**Keywords:** abdominal wall pain, ACNES, Carnett’s sign, consultation, undiagnosed

## Abstract

This study aimed to investigate consultation outcomes from gastroenterologists to generalist physicians for the diagnostic workup of undiagnosed chronic abdominal pain. This was a single-center, retrospective, descriptive study. Patients were included who were ≥15 years old and consulted from the Department of Gastroenterology to the Department of Diagnostic Medicine, to establish a diagnosis for chronic abdominal pain, at the Dokkyo University Hospital from 1 April 2016 to 31 August 2020. We retrospectively reviewed the patients’ medical charts and extracted data. A total of 12 cases were included. Eight patients (66.7%) were diagnosed with and treated for functional gastrointestinal disorders (FGID) at the Department of Gastroenterology; their lack of improvement under treatment for FGID was the reason for their referral to the Department of Diagnostic Medicine for further examination. After this consultation, new possible diagnoses were generated for eight patients (66.7%). Six of the eight patients (75.0%) were diagnosed with abdominal wall pain (anterior cutaneous nerve entrapment syndrome, *n* = 3; myofascial pain, *n* = 1; falciform pain, *n* = 1; and herpes zoster non-herpeticus; *n* = 1). Consultation referral from gastroenterologists to generalists could generate new possible diagnoses in approximately 70% of patients with undiagnosed chronic abdominal pain.

## 1. Introduction

Abdominal pain is one of the most frequent complaints for which patients seek medical attention. Although advances in diagnostic technology have made it possible to diagnose the causes of abdominal pain quickly and accurately, there are still patients who have undiagnosed chronic abdominal pain [1]. To obtain a diagnosis, these patients must visit multiple medical institutions, including tertiary hospitals. Gastroenterology departments in tertiary hospitals usually play a critical role in diagnosing difficult abdominal pain; detailed investigations can provide patients with definitive diagnoses. However, because abdominal pain can develop from causes other than gastrointestinal diseases, even after workup by gastroenterology specialists, some patients still require a diagnosis. Since the general internal medicine department is another department where the prevalence of referred patients with undiagnosed abdominal pain is also high [2], in such cases, a consulted generalist may assist diagnosis.

Abdominal wall pain (pain derived from the abdominal wall) is a key etiology in patients who have undiagnosed abdominal pain [3]. According to a past report, 10% to 30% of patients presenting to gastroenterologists with chronic abdominal pain were eventually diagnosed with chronic abdominal wall pain [4]. In addition, early recognition of abdominal wall pain has also been shown to reduce medical costs [5,6]. While the diagnosis of abdominal wall pain usually requires an only medical history, physical examination, and diagnostic treatment [7], some patients require extensive diagnostic intervention to rule out intra-abdominal organic diseases or functional gastrointestinal disorder (FGID) [3,8]. Therefore, workup by gastroenterologists is important before the diagnosis of abdominal wall pain in these patients. However, during the gastroenterologist’s diagnostic workup process, cognitive biases such as diagnostic momentum [9] and déformation professionnelle (the tendency to look at things from the perspective of one’s own profession or special expertise) [10] can prevent the physician from considering abdominal wall pain as a differential diagnosis. In such situations, generalist physicians can assist the specialist to consider abdominal wall pain as a cause of undiagnosed abdominal pain. Indeed, a previous study showed that abdominal wall pain was the most frequently developed diagnosis in patients referred to the general internal medicine department of the university hospital [2].

The characteristics of patients who are good candidates for consultation from gastroenterologists to generalists remain unknown. Therefore, we conducted this study to reveal the outcomes of consultation from gastroenterologist to generalist for diagnostic workup of undiagnosed abdominal pain, and also to reveal the demographics of abdominal pain, which gastroenterologists may wish to consult.

## 2. Materials and Methods

### 2.1. Study Design, Setting, and Subjects

This was a single-center, retrospective, descriptive study conducted at Dokkyo Medical University Hospital. Patients were included who were ≥15 years old and referred from the Department of Gastroenterology to the Department of Diagnostic and Generalist Medicine, for establishing a diagnosis of abdominal pain, from 1 April 2016 to 31 August 2020. Patients whose abdominal pain persisted less than two weeks or who had fever were excluded. The study was approved by the Independent Ethics Committee of the Dokkyo Medical University Hospital (No. R-41-10J). The requirement for obtaining informed consent was waived because this study was a retrospective data-only analysis.

### 2.2. Data Collection

We retrospectively reviewed the subjects’ medical charts and extracted data regarding sex, age at the time of the referral visit, time from the onset of illness to referral, number of institutions or departments visited before referral, details of the histories of abdominal pain (mode of onset, location, aggravating factors, relieving factors, radiation, quality of pain, severity, and accompanying symptoms), details of the examinations, and the therapeutic interventions performed at the Departments of Gastroenterology and Diagnostic and Generalist Medicine, respectively, the final diagnosis, and the prognosis. Details of the examinations and therapeutic interventions included abdominal tenderness (including location), Carnett’s sign, laboratory tests, abdominal plain radiography, abdominal computed tomography (CT), magnetic resonance cholangiopancreatography (MRCP), abdominal ultrasonography, upper and lower gastrointestinal endoscopies, and the type of therapy (drug, local injection, surgery). For confirmation of the diagnosis of abdominal wall pain, we used the criteria that abdominal wall pain diagnosis must include either a positive Carnett’s sign, localized extent of pain, or pain elicited only by certain movements, and that no other diagnostic test findings could explain the symptoms [5,11,12].

### 2.3. Statistical Analysis

Continuous data are presented as median (25th and 75th percentiles) or range. Categorical data are presented as counts and proportions (%). Continuous data were compared using the Mann–Whitney *U* test. Categorical data were compared using the chi-squared test or Fisher’s exact test. *p*-values < 0.05 were considered statistically significant. All statistical analyses were performed using R version 3.6.3 software (R Foundation for Statistical Computing, Vienna, Austria).

## 3. Results

### 3.1. Patient Backgrounds

A total of 12 cases were included (Figure 1). The median age was 67 years (36–72 years); seven (58.3%) subjects were male. The time from the onset of illness to referral to the Department of General Medicine ranged from two weeks to more than 10 years, with a median of 18 months (11–71 months). The median number of medical institutions or departments consulted before the referral to the Department of General Medicine was two (1.3–2) (Table 1). The most common site of abdominal pain was the left lower quadrant (*n* = 4), followed by epigastric (*n* = 3), left upper quadrant (*n* = 2), mid lower (*n* = 2), right upper quadrant (*n* = 1), and right flank (*n* = 1) (Figure 2). Only one patient had abdominal pain at two sites (epigastric and left lower quadrant).

The numbers indicated the case number in Table 1. 

### 3.2. Diagnostic Examinations and Diagnoses at the Department of Gastroenterology

Carnett’s sign was not documented in all cases (*n* = 0, 0.0%). Blood tests (*n* = 12, 100%), abdominal CT (*n* = 11, 91.7%), and upper gastrointestinal endoscopy (*n* = 10, 83.3%) were conducted in most of the patients. However, lower gastrointestinal endoscopy (*n* = 7, 58.3%), abdominal ultrasonography (*n* = 5, 41.7%), and MRCP (*n* = 2, 16.7%) were conducted selectively. Eight patients (66.7%) were diagnosed and treated as FGID (functional dyspepsia, *n* = 7; irritable bowel syndrome, *n* = 1). In all eight patients, lack of improvement under treatment for FGID was the reason for their referral to the Department of General Medicine.

### 3.3. Diagnostic Examinations and Diagnoses at the Department of Diagnostic and Gereralists Medicine

Carnett’s sign was documented in six cases (50%). Additional diagnostic tests were performed in only four cases (33.3%; blood test, *n* = 1; spinal MRI, *n* = 3). In total, new possible diagnoses were generated in eight patients (66.7%). In these patients, six (75.0%) were diagnosed as abdominal wall pain (anterior cutaneous nerve entrapment syndrome [ACNES], *n* = 3; myofascial pain, *n* = 1; falciform pain, *n* = 1; herpes zoster non-herpeticus, *n* = 1). Of the other patients, one was diagnosed as chronic pelvic pain, and the other as iron deficiency. In patients with abdominal wall pain, only one (16.7%) was aware of localized pain; the other five (83.3%) complained of relatively widespread pain. None of the patients with abdominal wall pain had weight loss; all of them had localized (not multiple or diffuse) tenderness in the abdomen.

### 3.4. Treatments and Follow Up

The treatments provided were as follows: exercise, *n* = 1; increase calorie intake, *n* = 1; acetaminophen, *n* = 3; non-steroidal anti-inflammatory drugs (NSAIDs), *n* = 2; eperisone, *n* = 1; pregabalin, *n* = 1; tender point injection, *n* = 2; neurectomy, *n* = 1; iron supplementation, *n* = 1, and magnesium oxide, *n* = 1. In the nine cases (75%) that were followed-up (median duration, 5 weeks), symptoms improved in 2 (1 ACNES and 1 iron deficiency).

### 3.5. Abdominal Wall Pain vs. Nonabdominal Wall Pain

There were no statistically significant differences between patients with and without abdominal wall pain (Table 2).

## 4. Discussion

In this study, we found that new possible diagnoses were generated in 66.7% of 12 patients who were referred from gastroenterologists to generalists for undiagnosed chronic abdominal pain in the setting of a tertiary hospital. Before referral to the generalists, 66.7% of patients were diagnosed as FGID. In the patients who obtained new possible diagnoses after referral to the generalists, abdominal wall pain accounted for 75%; ACNES was the most common diagnosis.

The prevalence of abdominal wall pain in the study (50%) was higher than that has been reported in previous studies (15–28%) [13,14,15]. This could be explained by differences in the subjects’ characteristics. Previous studies included patients who required emergency hospitalization with intra-abdominal etiology, patients with abdominal pain who were referred to the pain clinic after the exclusion of intra-abdominal disease, patients who had been referred to the hospital by a general practitioner, and those who required hospitalization for an unknown cause. In contrast, our study included only patients who were referred by physicians of the Department of Gastroenterology at a university hospital, who had highly-developed skills that enabled them to exclude acute and chronic intra-abdominal organic diseases. Thus, there could be a high prevalence of abdominal wall pain.

Chronic abdominal pain is often treated as functional pain because of the absence of organic abnormalities in various tests [8]. In this study, eight patients (66.7%) were referred to the generalists because of a lack of improvement of symptoms under treatment for FGID. Six of these patients obtained additional possible diagnoses from the generalists, including three cases of abdominal wall pain. As previous studies have suggested, abdominal wall pain should be considered in patients with refractory abdominal pain and suspected FGID.

Abdominal wall pain may present with atypical patterns. Among the patients in our study, the typical presentation included the duration of symptoms and the region of pain. Abdominal wall pain lasting longer than one year is reported to be common [5]. In this study, the duration of illness seemed to be longer in patients with abdominal wall pain compared with those without it, though the difference was not statistically significant. Regarding the region of pain, all patients in our study who complained of right-sided pain were diagnosed with abdominal wall pain. This finding is consistent with previous reports [1,3]. The atypical patterns of abdominal wall pain observed in this study were with age and area of pain. The most common age group for abdominal wall pain is 30–50 years [16,17]. However, in this study, the median age of patients diagnosed with abdominal wall pain was 70 years. It is noteworthy that although localized pain is recognized as the typical pattern in patients with abdominal wall pain, 83.3% of patients with abdominal wall pain in this study complained of relatively widespread abdominal pain (compared with the area of tenderness on examination). These two atypical presentations could be reasons for the difficulty of diagnosis of abdominal wall pain. Abdominal wall pain can occur at any age [12,16], and relatively widespread abdominal pain can also occur in patients with abdominal wall pain [11,18]. Therefore, even in patients with atypical presentation, abdominal wall pain should be considered when patients are referred by gastroenterologists to generalists for undiagnosed abdominal pain.

Carnett’s sign is an important physical examination for the diagnosis of abdominal wall pain [1]. Carnett’s sign is usually defined as positive when the abdominal tenderness does not change or increases when a patient raises the head or legs in the supine position while an examiner presses the point of maximal abdominal tenderness [19]. With the definition, a positive Carnett’s sign indicates that a cause of abdominal pain exists in the abdominal wall, such as nerve entrapment syndromes or myofascial pain [20]. Carnett’s sign is recognized as a useful examination for detecting abdominal wall pain: a study reported that Carnett’s sign showed the sensitivity of 78% and the specificity of 88% [3], and another study also reported that positive likelihood ratio of Carnett’s sign was 2.6 [21]. In addition, Carnett’s sign may help discriminate psychogenic abdominal pain [21]. Therefore, Carnett’s sign should be included in the diagnostic workup of abdominal pain.

However, for patients with chronic abdominal pain, it may not be tested as much as might be expected in clinical practice. Carnett’s sign was not documented in all patients by the gastroenterologists; it was documented by the generalists in only 50% of the patients. This might be explained by the fact that abdominal wall pain remains unfamiliar to physicians, as it is rarely described in detail in textbooks [4,5]. Also, physicians may not consider the possibility of abdominal wall pain when seeing patients who complain of widespread abdominal pain. Yet, as shown in our study, a localized tenderness and/or positive Carnett’s sign can be detected in these patients. Therefore, performing Carnett’s sign should be encouraged for patients with chronic abdominal pain, even when their pain seems to be atypical.

Early diagnosis of abdominal wall pain by eliciting Carnett’s sign can prevent unnecessary diagnostic journeys such as overtesting, frequent clinical visits, and referral, and also can prevent adverse events from ineffective treatment [22,23]. However, Carnett’s sign should be interpreted with caution in patients with unexplained weight loss or tenderness at multiple points who have not been investigated for intra-abdominal diseases, since they have a risk of visceral organic diseases [5,11]. Visceral organic diseases seemed to be appropriately excluded by gastroenterologists. None of this study’s cases which led to the diagnosis of abdominal wall pain had weight loss and none had multiple tenderness in the abdomen. Therefore, the reliability of a positive Carnett’s sign could be confirmed in this study.

Trigger point injection is another valuable diagnostic investigation for the diagnosis of chronic abdominal pain. In general, the trigger point injection test is positive when abdominal pain improves by anesthetic injection to the point of tenderness, and the positive result of trigger point injection indicates the diagnosis of abdominal wall pain [3,12,24]. While the diagnostic value of trigger point injection for abdominal pain can be maximized with the positive Carnett’s sign [16], trigger point injection can also be used for the diagnosis of abdominal wall pain even in patients who are suspected with abdominal wall pain but negative for Carnett’s [3,12]. In our study, trigger point injection was conducted only in two patients. Among four patients who were not diagnosed even after the consultation to the Department of Diagnostic and Generalist Medicine in our study, trigger point injection would have helped to detect the diagnosis.

This study has several limitations. First, since this was a small retrospective study conducted at a single center, the results should be used only for research and hypothesis generation; they should not be considered as generalizable evidence. In particular, the result of the analyses did not provide any significant difference, which seemed mainly due to the small sample size. Second, since definitive confirmation of the abdominal wall pain diagnosis is difficult, and Carnett’s sign was not documented in 50% of the patients in this study, the diagnosis could be wrong in some patients. Third, since not all patients with undiagnosed chronic abdominal pain were referred from gastroenterologists to generalists, selection bias could not be avoided.

There seemed several reasons that relatively small participants (*n* = 12) were included for around 4.5 years in this study. First, some physicians did not have the idea that consultation to the Department of Diagnostic and Generalist Medicine may be an option for detecting the cause of chronic abdominal pain. Indeed, to the best of our knowledge, no previous study reported the results of diagnostic consultation from the gastroenterology department to the general medicine department for patients with chronic abdominal pain. The absence of such a study would indirectly support the possibility that diagnostic consultation from gastroenterology departments to general medicine departments is not routinely performed. Second, only the most diagnostically difficult cases that require well-honed medical history and examination skills by general physicians after a detailed investigation in the Department of Gastroenterology might have been referred for consultation to the Department of Diagnostic and Generalist Medicine. Typical cases of abdominal wall pain seemed not to be consulted because gastroenterologists could diagnose them. This may have resulted in a decrease in the number of participating cases. Third, some patients were referred directly from a clinic or a hospital to the Department of Diagnostic and Generalist Medicine, not via the Department of Gastroenterology, when the physician in the clinic or hospital thought that other than gastroenterology disease was a cause of abdominal pain. These cases were not included in the inclusion criteria of this study, but they could have been originally referred to the general medicine department after seeing the gastroenterology department once.

## 5. Conclusions

Consultation from gastroenterologists to generalists in the tertiary hospital could generate new possible diagnoses in around 70% of patients with undiagnosed chronic abdominal pain. Patients with chronic abdominal pain diagnosed as FGID could be indicated for referral from gastroenterologists to generalists in order to seek alternative diagnoses. Abdominal wall pain should be considered as a diagnosis even in elderly patients who have relatively widespread pain; physicians should perform Carnett’s sign in these cases.

## Figures and Tables

**Figure 1 healthcare-09-01150-f001:**
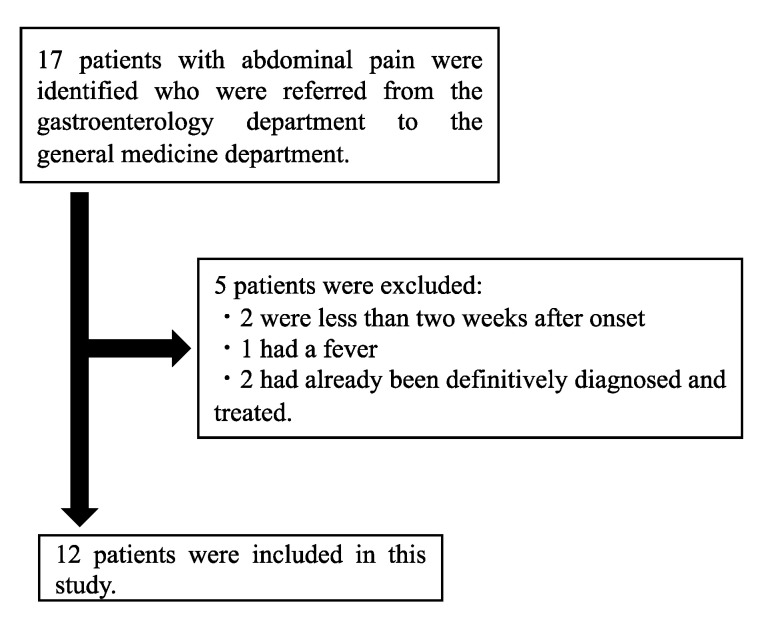
Flow diagram for the inclusion of patients in this study.

**Figure 2 healthcare-09-01150-f002:**
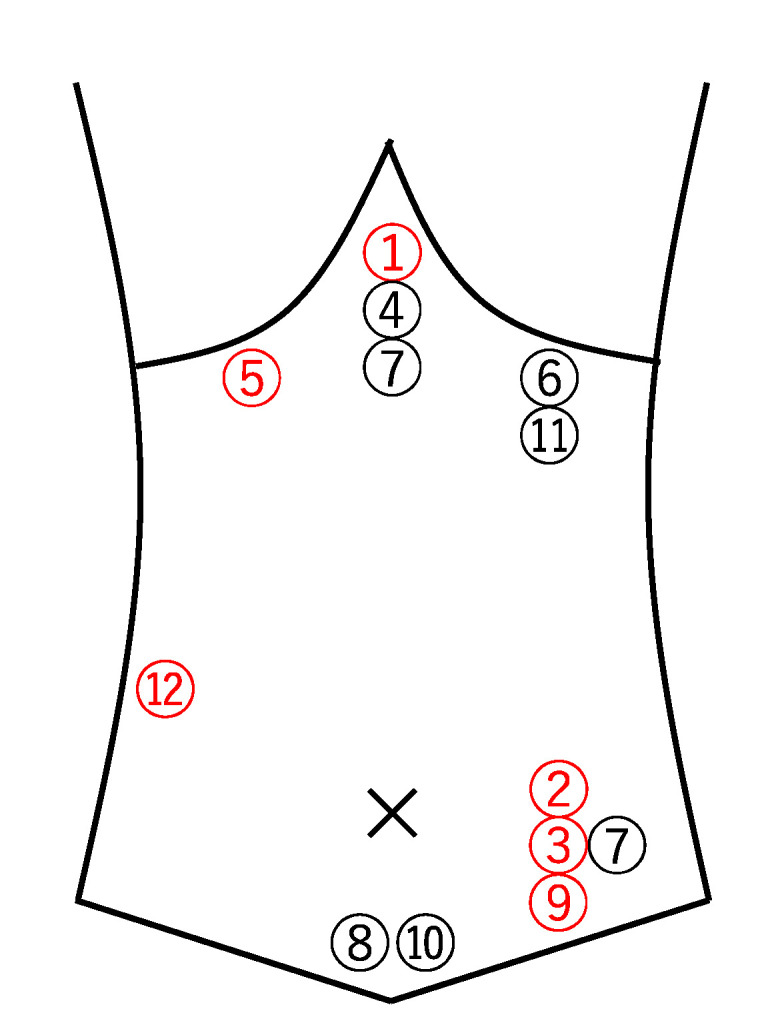
The sites of abdominal pain. (Red number indicate abdominal wall pain).

**Table 1 healthcare-09-01150-t001:** Twelve patients with undiagnosed abdominal pain.

Case #Age Sex	Duration of Symptoms	Number of Medical Institutions	Presenting Features	Diagnostic Workups, Working Diagnoses, and Treatments before Referral to Generalists	Additional Diagnostic Workups, Revised Diagnosis, Treatments, and Outcome after Referral to Generalists
1 71 F	More than 10 years	3	Epigastric pain with back pain. Not related to eating or posture. Tenderness of the xiphoid process.	**Workups:** Labs, CT, EGD **Diagnosis:** Functional dyspepsia **Treatments:** Acotiamide hydrochloride hydrate, Chinese herbal medicine (Rikkunshito)	**Additional workups:** None **Diagnosis:** Xiphoidynia **Treatments:** NSAIDs **Outcome:** Symptoms did not improve during 6-week follow up
2 66 F	1 year	2	Left flank pain with back pain. Not related to eating or posture. Tenderness of lateral edge of the left rectus abdominis and left erector spinae. Positive Carnett’s sign.	**Workups:** Labs, CT (twice), MRCP (twice), ultrasound (twice), EGD, colonoscopy **Diagnosis:** functional dyspepsia **Treatments:** Acotiamide hydrochloride hydrate	**Additional workups:** Thoracic spinal MRI **Diagnosis:** ACNES. **Treatments:** Acetaminophen, NSAIDs, eperisone hydrochloride, tender point infiltrations. **Outcome:** Symptoms did not improve during 5-week follow up
3 86 F	10 years	Unidentified	Left lower quadrant pain. Exacerbated by getting up and straining. Very localized tenderness in the left lower quadrant. Positive Carnett’s sign.	**Workups:** Labs, CT, MRCP, EGD (twice), colonoscopy (twice) **Diagnosis:** None **Treatments:** Acetaminophen, NSAIDs, pregabalin	**Additional workups:** None **Diagnosis:** ACNES **Treatments:** Exercise **Outcome:** No follow up
4 73 M	2 years	Unidentified	Epigastric squeezing pain. No tenderness. Not related to eating. Negative Carnett’s sign.	**Workups:** Labs, CT, EGD, colonoscopy **Diagnosis:** None **Treatments:** Laxatives	**Additional workups:** Lumber spinal MRI **Diagnosis:** None **Treatments:** None **Outcome:** Symptom persisted for 3 years
5 69 M	1 year	2	Right upper quadrant pain. Induced by twisting the body to the left.	**Workups:** Labs, ultrasound, CT, EGD **Diagnosis:** Functional dyspepsia **Treatments:** Acotiamide hydrochloride hydrate, esomeprazole, magnesium hydrate	**Additional workups:** None **Diagnosis:** Myofascial pain **Treatments:** None **Outcome:** No follow up
6 45 F	1 year	1	Left upper quadrant pain with headache and stiff shoulders. Exacerbated by prolonged sitting. Tenderness in left 12th rib.	**Workups:** Labs, ultrasound, EGD **Diagnosis:** Functional dyspepsia, GERD **Treatments:** Acotiamide hydrochloride hydrate, Chinese herbal medicine (Rikkunshito), vonoprazan fumarate	**Additional workups:** None **Diagnosis:** Iron deficiency **Treatments:** Iron supplementation **Outcome:** Symptoms improved after 1 month
7 19 M	2–3 years	1	Epigastric and left lower quadrant pain with tenderness. Gradual onset. Not related to eating. Pain did not occur in the supine position. Negative Carnett’s sign.	**Workups:** Labs, ultrasound, CT, EGD **Diagnosis:** Functional dyspepsia **Treatments:** Acotiamide hydrochloride hydrate, esomeprazole, magnesium hydrate, dimethicone	**Additional workups:** None **Diagnosis:** None **Treatments:** None **Outcome:** Symptoms did not improve during 2-week follow up
8 68 F	4.5 years	2	Lower abdominal dull pain with tenderness. Gradual onset.	**Workups:** Labs, ultrasound, CT, EGD, colonoscopy, PET-CT **Diagnosis:** Functional dyspepsia **Treatments:** Acotiamide hydrochloride hydrate, esomeprazole, magnesium hydrate	**Additional workups:** None **Diagnosis:** Chronic pelvic pain **Treatments:** Pregabalin and magnesium oxide **Outcome:** No visits after 1-year follow up
9 76 M	10 years	1	Left abdominal pain. Not related to eating or posture. Tenderness 5 cm outside the navel. Positive Carnett’s sign.	**Workups:** Labs, CT, EGD, colonoscopy **Diagnosis:** None **Treatments:** None	**Additional workups:** Cervical and thoracic spinal MRI **Diagnosis:** ACNES. **Treatments:** Tender point infiltrations and neurectomy **Outcome:** Initial symptom improved but some other pain occurred
10 15 M	7 months	2–3	Lower abdominal pain with tenderness. Acute onset. Not related to eating or posture. No tenderness.	**Workups:** Labs, CT, colonoscopy **Diagnosis:** Irritable bowel syndrome **Treatments:** NSAIDs, tramadol hydrochloride, acetaminophen, ramosetron hydrochloride, escitalopram oxalate, sulpiride	**Additional workups:** Labs **Diagnosis:** None **Treatments:** Acetaminophen **Outcome:** Symptoms did not improve during 3-week follow up
11 23 M	3 months	2	Left upper quadrant pain with left back pain and heartburn. Acute onset. Exacerbated by eating.	**Workups:** Labs, abdominal X-ray, CT, EGD, colonoscopy **Diagnosis:** GERD, functional dyspepsia, irritable bowel syndrome **Treatments:** Vonoprazan fumarate, acotiamide hydrochloride hydrate, trimebutine maleate	**Additional workups:** None **Diagnosis:** None (suspicious of superior mesenteric artery syndrome) **Treatments:** Increasing calorie intake **Outcome:** No visits after 1-month follow up
12 40 M	2 weeks	2	Right flank tingling pain to the back. Acute onset. Not related to eating.No skin rash. Negative Carnett’s sign.	**Workups:** Labs, CT **Diagnosis:** None **Treatments:** Acetaminophen and pregabalin	**Additional workups:** None **Diagnosis:** Herpes zoster **Treatments:** Increasing the dose of acetaminophen. **Outcome:** No follow up

**Table 2 healthcare-09-01150-t002:** Twelve patients with undiagnosed abdominal pain.

	Abdominal Wall Pain	Nonabdominal Wall Pain	*p* Value *
Age (years)	70 (66.75–74.75)	34 (20–62.25)	0.09
Male	3/6 (50%)	4/6 (66.7%)	>0.99
Period of illness (months)	66 (12–120)	18 (8.25–28.5)	0.46
Number of visited medical institutions	2 (2–2)	2 (1–2)	0.40
Characteristics of abdominal pain			
Related to eat	Yes: 0/6 (0%) No: 4/6 (66.7%) Unknown: 2/6 (33.3%)	Yes: 1/6 (16.7%) No: 3/6 (50.0%) Unknown: 2/6 (33.3%)	>0.99
Related to postures	Yes: 2/6 (33.3%) No: 3/6 (50.0%) Unknown: 1/6 (16.7%)	Yes: 2/6 (33.3%) No: 1/6 (16.7%) Unknown: 3/6 (50.0%)	0.77
**Tests conducted**			
Labs	6/6 (100%)	6/6 (100%)	
Ultrasound	2/6 (33.3%)	3/6 (50.0%)	
Abdominal X-ray	0/6 (0%)	1/6 (16.7%)	
MRCP	2/6 (33.3%)	0/6 (0%)	
EGD	5/6 (83.3%)	5/6 (83.3%)	
Colonoscopy	3/6 (50.0%)	4/6 (66.7%)	
**Tentative diagnosis before referral**			0.55
Functional dyspepsia	3/6 (50.0%)	4/6 (66.7%)	
GERD	0/6 (0%)	2/6 (33.3%)	
Irritable bowel syndrome	0/6 (0%)	2/6 (33.3%)	
None	3/6 (50.0%)	0/6 (0%)	
**Carnett’s sign**			0.32
Positive	3/6 (50.0%)	0/6 (0%)	
Negative	1/6 (16.7%)	2/6 (33.3%)	
Unknown	2/6 (33.3%)	4/6 (66.7%)	
**Final diagnosis**	ACNES: 3/6 (50.0%) Myofascial pain: 1/6 (16.7%) Xiphoidynia: 1/6 (16.7%) Herpes zoster: 1/6 (16.7%)	Chronic pelvic pain: 1/6 (16.7%) Iron deficiency: 1/6 (16.7%) Unknown: 4/6 (66.7%)	

* *p* values were generated by the Mann–Whitney *U* test or Fisher’s exact test.

## Data Availability

The data sets used and analyzed during the current study are available from the corresponding author on reasonable request.

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
