# Peer review of "Diagnostic Effect of Consultation Referral from Gastroenterologists to Generalists in Patients with Undiagnosed Chronic Abdominal Pain: A Retrospective Study"

_healthcare, 2021, doi:10.3390/healthcare9091150_

Round 1

Reviewer 1 Report

The introductory section must be developed and  more recent references should be added. 

Results section: raws 89-91 should be removed. 

Limitations of the study: the low number of patients did not provide any statistically significant result (all values of p are above 0.05). 

The number of patients is low in a quite log period of time. Please explain or add some more cases if possible (preferred). 

Discussion section: it has to be developed and more references should be added. 

Raws 185-186: too many references are indexed after a single sentence ([1–7,10,11,13–16]). Develop/detail the sentence. 

Reviewer 2 Report

The authors present an interesting study regarding the consultation of gastroenterologist for diagnosis of abdominal pain.

I have no concerns of the manuscripts content. However, I have few questions/possible corrections. Please find my comments below.

3. Results (rows 89–91): "This section may be divided by subheadings. It should provide a concise and precise description of the experimental results, their interpretation, as well as the experimental conclusions that can be drawn." This is probably copied from the author guidelines?

Results 3.2 (row 109): "Carnett’s sign was not documented in all cases." Could this sentence be modified to provide the n and % of the documented Carnett's sign in a similar manner to following sentences?

Table 2. Male gender. Is the p-value really 1.0?

Round 2

Reviewer 1 Report

The authors addressed all the required changes and suggestions. Thus, the value of the manuscript increased. I recommend publication in this form.